# An Empirical Study of Adder Neural Networks for Object Detection

**Xinghao Chen**[1], **Chang Xu**[2], **Minjing Dong**[2,1], **Chunjing Xu**[1], **Yunhe Wang**[1*]
[1]Huawei Noah's Ark Lab
[2]School of Computer Science, University of Sydney
{xinghao.chen, yunhe.wang}@huawei.com, c.xu@sydney.edu.au

## Abstract

Adder neural networks (AdderNets) have shown impressive performance on image classification with only addition operations, which are more energy efficient than traditional convolutional neural networks built with multiplications. Compared with classification, there is a strong demand on reducing the energy consumption of modern object detectors via AdderNets for real-world applications such as autonomous driving and face detection. In this paper, we present an empirical study of AdderNets for object detection. We first reveal that the batch normalization statistics in the pre-trained adder backbone should not be frozen, since the relatively large feature variance of AdderNets. Moreover, we insert more shortcut connections in the neck part and design a new feature fusion architecture for avoiding the sparse features of adder layers. We present extensive ablation studies to explore several design choices of adder detectors. Comparisons with state-of-the-arts are conducted on COCO and PASCAL VOC benchmarks. Specifically, the proposed Adder FCOS achieves a 37.8% AP on the COCO val set, demonstrating comparable performance to that of the convolutional counterpart with an about $1.4\times$ energy reduction.

## 1 Introduction

Object detection is a foundational problem in computer vision and has attracted tremendous interests from both academic and industrial communities for decades [21]. It has a wide range of applications for various areas, *e.g.*, video surveillance, autonomous driving and robotic vision. Deep neural networks have indeed dominated the research of object detection in recent years since the pioneering work of R-CNN [12]. The performance of object detectors has been considerably improved by various designs of architecture, loss function *etc*. However, most modern accurate object detectors require massive computation, making them quite challenging in resource-constraint applications, *e.g.*, mobile phones and embedded devices.

Various approaches have been proposed to compress and accelerate convolutional neural networks (CNNs) for classification tasks, including channel pruning [24, 14, 26, 38], low-bit quantization [50, 25] and lightweight network design [33, 27]. These methods reduce the number of parameters or inference latency while maintaining the accuracy to the maximum extent. Such model compression methods have also been explored in a variety of down-stream tasks, such as semantic segmentation [7] and image super resolution [44] *etc*.

There have been a few methods aiming at fast and efficient object detectors. One family of solutions is using new architecture design [23, 31, 29, 41]. For example, YOLO series [30, 31] have achieved good trade-off between running speed and accuracy via a novel one-stage detection framework. Another family of solutions is to use common model compression methods for accelerating object

---

[*]Corresponding author.

35th Conference on Neural Information Processing Systems (NeurIPS 2021).

detectors, *e.g.*, knowledge distillation [4, 42] and pruning [1]. Moreover, some recent works utilize neural architecture search (NAS) approach for searching better architectures for different components of object detectors [3, 13, 11]. Although these methods mentioned above show strong performance while improving the efficiency, they are mainly built with traditional convolutional neural networks, which contain massive inefficient multiplications.

Recently, Chen *et al.* [5] proposed the adder neural networks (AdderNets) to replace traditional convolutional filters with adder filters. Since addition is more energy efficient than multiplication [36, 47], AdderNets shed light on the design of efficient neural networks and have the potential of much fewer chip areas and less energy consumption. AdderNets have shown impressive performance in large scale image classification [45] via a kernel-based progressive distillation method, and also successfully been applied for other applications like image super resolution. Song *et al.* [34] proposed to utilize self shortcuts and learnable power activations to build super resolution networks via adder filters.

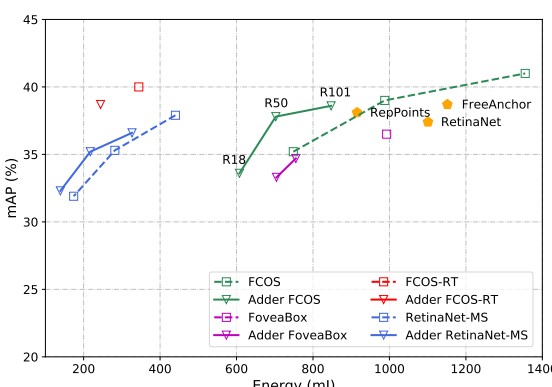

Figure 1: Comparisons of mAP on COCO *val2017* and energy costs for different object detectors.

Existing variants of AdderNets mainly deal with either image classification [5, 45, 9] or super-resolution tasks [34]. It is not clear yet how will the AdderNets perform for object detection, which often has sophisticated framework design and various objectives. This much more challenging computer vision task therefore brings in new challenges and opportunities for the research on AdderNets. *So how to build accurate and efficient object detectors via AdderNets?* The straightforward idea is to directly replace the original convolutional filters by adder filters. However, this naive adder detector cannot be easily trained as classification networks. At first, most modern detectors tend to pre-train the backbone on the ImageNet and then fine-tune the whole model on the target dataset, but this straightforward fine-tuning might worsen adder detectors, because of the sensitivity of adder filter. Moreover, the performance-proven neural architectures were all developed for convolution based detectors, and whether they are still applicable for the adder detectors is unclear.

In this paper, we propose a series of strategies to reform efficient object detectors with adder filters. In contrast with the frozen batch normalization widely exploited during fine-tuning of the detector, we empirically observe an opposite conclusion that adder detectors are better to unlock the statistics of batch normalization in pre-trained adder backbone for a performance improvement. Extensive ablation studies are conducted to explore the properties of batch normalization layers and the impact of batch size. In addition, a new feature fusion network with more residual connections and a better fusion module is explored to compensate for sparse adder features. Experimental results are reported on PASCAL VOC and COCO benchmarks, and the results are carefully analyzed and discussed. In particular, the proposed Adder FCOS achieves a 37.8% mAP on COCO val set, which is comparable with state-of-the-art object detectors, while saving much energy consumption as shown in Figure 1. In summary, we present an extensive empirical study for how to build object detectors via adder neural networks. We believe that the discussions and analysis in this paper will be beneficial for the research of efficient object detection and adder neural networks.

## 2 Related Work

**Object Detection.** Object detection has attracted tremendous interests for decades. Although there have been huge improvements for accurate object detection [21], the struggle for high-speed and energy-efficient detectors is largely unsolved. Several recent approaches exploit neural architecture search (NAS) to discover efficient and accurate object detectors [3, 13, 11]. On the meanwhile, there are some attempts to utilize model compression methods for improving the efficiency of detectors, *e.g.*,channel pruning [1], quantization [43] and knowledge distillation [4, 42]. Nevertheless, convolutional neural networks (CNNs) have in fact dominated the design of object detectors, which consist of massive multiplications and are quite energy inefficient.

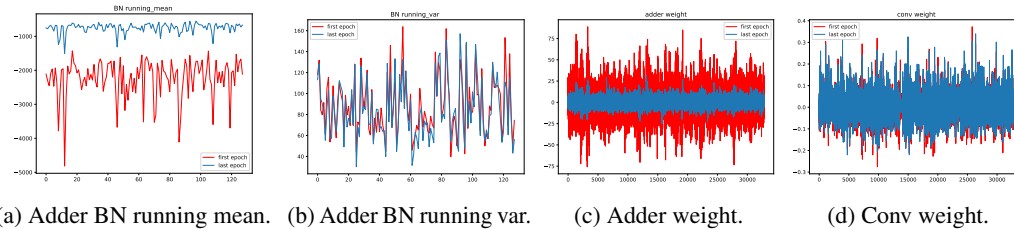

| (a) Adder BN running mean. | (b) Adder BN running var. | (c) Adder weight. | (d) Conv weight. |

Figure 2: The statistics of batch normalization and filter weights in backbone during training.

**Adder Neural Networks.** Chen *et al.* [5] provided a new perspective for designing energy efficient neural networks. They proposed to utilize $\ell_1$ distance to measure the similarity between the input features and filter weights. The devised AdderNets consist of only additions instead of massive multiplication, which are potentially energy efficient. Full precision gradients along with adaptive local learning rate are exploited to optimize this novel neural networks. AdderNets show impressive performance on large-scale image classification benchmark, *i.e.*, ImageNet. To further improve the performance of AdderNets, Xu *et al.* [45] proposed a kernel based knowledge distillation method to learn from homogeneous CNNs. Using this method, AdderNets achieve even better accuracy than ResNet-50 on ImageNet. You *et al.* [47] further proposed to use bit-shift operations and additions to improve the energy efficiency. Song *et al.* [34] analyzed several key problems of AdderNets and successfully applied AdderNets for image super-resolution. Nevertheless, designing accurate object detectors with AdderNets is still less explored and is the main focus of this paper.

## 3 AdderNets for Object Detection

Traditional deep convolutional neural networks are mainly constructed by convolutional filters, which have massive manipulations and are energy inefficient. To this end, Chen *et al.* [5] proposed a new kind of neural architecture called AdderNet, which adopts $\ell_1$-norm to compute the similarity between the input features $X$ and the filter weights $F$, as shown in Eq. (1).

$$Y(m,n,t) = -\sum_{i=0}^{d}\sum_{j=0}^{d}\sum_{k=0}^{c_{in}} |X(m+i, n+j, k) - F(i,j,k,t)|. \tag{1}$$

We aim to extend the success of AdderNet to the object detection task. We build our adder object detectors upon modern convolution-based detection frameworks, *e.g.*, FCOS [39]. It is straightforward to replace the convolutional filters in the detector with adder filters. However, it is non-trivial to train an adder detector of decent performance. We first analyze several key strategies for applying adder filters for detectors. We then propose a novel multi-scale feature fusion architecture which is more suitable for adder detectors.

### 3.1 Making it Work: Towards a Strong Baseline

Intuitively if we replace the convolutional filters with adder filters in modern object detectors like FCOS [39], we could obtain a vanilla adder detector, which may be denoted as Adder FCOS. However, it is not easy to train the vanilla Adder FCOS in the same way as convolutional FCOS. Here we elaborate on some special designs for adder detectors.

### 3.1.1 Revisiting Batch Normalization in Adder Detector

Most modern object detection methods follow the paradigm of pre-training the backbone network on large scale image classification dataset (*e.g.*, ImageNet) and then fine-tuning the whole detector on the target detection dataset. In object detection, the batch size is usually much smaller than that of image classification tasks, due to relatively higher resolution of the input images. For example, generally there are only 2 or 4 images on one GPU for training an modern object detector. Therefore, the statistics of batch normalization (BN) are often frozen (denoted as FrozenBN) in the fine-tuning stage, which brings in a considerable improvement than the unfrozen counterpart (simply denoted as BN here) [6]. However, we empirically observe that FrozenBN in backbone leads to an unstable training of adder detectors. As shown in Fig. 3, with FrozenBN in backbone, the training loss converges

much slower than the unfrozen BN, resulting in a quite unsatisfied performance with mAP of zero. Therefore, updating the statistics of BN layers in backbone network is critical for training a detector built with adder filters.

We attribute this phenomenon to the variance of features in AdderNets. As analyzed in [5], the addition operations in AdderNets tend to have much larger variances for the features before batch normalization. Dong *et al.* [10] also analyze that the mean and variance of output feature are dominated by the variance of adder weights. Therefore, slightly tuning the adder filters would bring in drastic changing of the feature distribution, which makes the previous statistic of batch normalization incompatible with the input features. We visualize the statistics of batch normalization for a random layer in backbone during training in Fig. 2. We can see that the weights for adder network (Fig. 2c) and convolutional network (Fig. 2d) exhibit quite different properties. The weights for convolutional network only have slight changes from the first epoch to the last epoch. How-

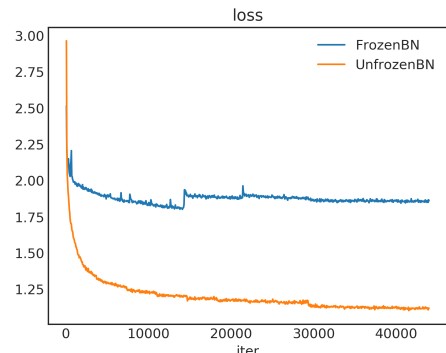

Figure 3: Training loss of fine-tuning adder detectors with frozen BN (final mAP: 0%) and unfrozen BN (final mAP: 32.4%).

ever, the weights for adder detector become much smaller as the training goes on. The changing of adder weights brings drastic variance for the output features, thus making the running means of BN for the last epoch quite different from that of the first epoch. If the statistics of BN are frozen during training, it would be quite challenging for training.

To address this problem, it is necessary to unfreeze the statistic of batch normalization when fine-tuning the adder detectors from a pre-trained backbone. However, recomputing statistics for BN layers would be critical for the final performance, especially when the batch size is small, *i.e.*, 2 images per GPU. Therefore, it is necessary to use a larger batch size for training adder detectors.

### 3.1.2 Better Pre-trained Backbone and Gradients

The performance of vanilla AdderNet [5] on ImageNet is still poorer than CNNs. For better performance, we use the pre-trained model using the knowledge distillation proposed in [45]. The Top-1 accuracy of AdderNet-50 on ImageNet is 76.8%, which is comparable with the performance of its convolutional counterpart ResNet-50. More discussions for the pre-trained models will be included in the ablation studies.

The original AdderNets [5] require a special design of gradient descent rule for optimization. More specifically, instead of using the actual sign gradients, AdderNets exploit the full precision gradients of $\ell_2$-norm to update the filters. To avoid the magnitude accumulation brought by the gradient chain rule, AdderNets clip the gradients of input features $X$ to $[-1, 1]$ using a HardTanh function. Though this practice works well in image classification tasks, we argue that this gradient approximation may introduce challenges for optimization for object detection which often exhibits more complicated architectures. Since the gradients of $X$ only play the role of accumulate gradients in chain rule, it is better to use the sign gradients for $X$. Therefore, we exploit the gradients of $\ell_2$-norm for adder weights and sign gradients for the input features.

### 3.2 Better Feature Fusion for Adder Detector

Using all strategies discussed in the above sections, we obtain a baseline Adder FCOS with the mAP of 34.8%, which is quite close to the convolutional counterpart. We then move one step forward to explore a better architecture for adder detectors.

We first visualize features from the pretrained backbone, as shown in Fig. 4. The feature maps from the last block of AdderNet-50 is much sparser than ResNet-50. More specifically, over 92% of the features for the last output from AdderNet-50 are zeros, while the percentage for ResNet-50 is only 63%. These sparse features may be enough for image classification task. However, it poses great challenges for object detection which needs dense predictions for class classification and bounding box regression.

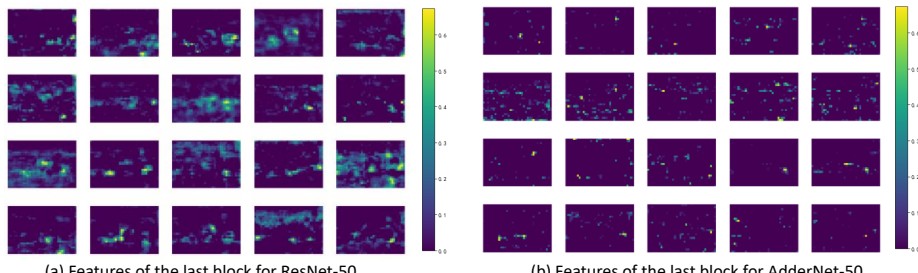

(a) Features of the last block for ResNet-50          (b) Features of the last block for AdderNet-50

Figure 4: Features of the last block for ResNet-50 and AdderNet-50. The features from AdderNet are much sparser than CNNs and pose great challenges for object detection which needs dense predictions for bounding box regression and category classification.

We attribute the problem of sparse features to the calculation of adder filters. As shown in Eq. (1), the output features of adder operation are always negative. Although the normalization procedure in BN makes these features have the mean of zero and the variance of one, the scaling and shift parameters in BN are learned to restore the representation power of the original features and tends to move features towards negative. The following ReLU activation would eliminate the negative features, making features in deeper layers to be more sparse.

Multi-scale feature fusion module is widely adopted for accurate object detection [19, 16, 22, 11]. As shown in Fig. 5 (a), feature Pyramid Network (FPN) [19] utilizes a top-down architecture to aggregate features from different levels and enhance the high-level semantic features for all scales. PAFPN [22] added extra bottom-up path for better feature aggregation as shown in Fig. 5 (b). The feature fusion module somehow alleviates the feature sparsity problem of adder detectors. However, since the pre-trained model of AdderNet has sparse features in top layers, the top-down path may not bring meaningful information for feature fusion.

Therefore, we hereby propose a novel feature fusion module to alleviate this problem. Unlike most previous multi-scale feature fusion methods [19, 22, 37] that first adopt top-down path for feature aggregation, we propose to exploit a reverse pattern. Specifically, the proposed R-PAFPN first exploits bottom-up path to propagate features from bottom layers to top layers, and then utilizes top-down path for fusing semantically strong features. This simple yet effective design is more suitable for adder detectors. Moreover, as discussed in [34], identity mapping is challenging for AdderNets. Therefore, we also add extra skip connections in our proposed feature fusion module. Specifically, we add residual connections for each 3x3 adder filters for better feature propagation, as shown in Fig. 5.

## 4   Experiments

In this section we first conduct extensive ablation experiments to analyze the effectiveness of different components of the proposed method. After that, we compare our proposed method with state-of-the-art object detectors.

### 4.1   Experimental Settings

We conduct experiments on the bounding box detection track of MS COCO 2017 and PASCAL VOC benchmarks, which have 80 and 20 object classes, respectively. On all experiments, the AdderNet backbone keeps the first layer as convolution and has all the rest layers built with adder filters, following the practice in [5, 45].

**COCO.** Following the common practices, for COCO benchmark we use the COCO *train2017* split that contains $118k$ images for training, *val2017* split for validation ($5k$) and *test-dev* split ($20k$) for testing. We report the average precision (AP) *w.r.t.* different IoU thresholds and different object scales, *i.e.*,mAP, $AP_{50}$, $AP_{75}$, $AP_S$, $AP_M$ and $AP_L$. All models are trained with stochastic gradient descent (SGD) over 8 GPUs. Unless otherwise specified, all models are trained for 12 epochs (also known as $1\times$ schedule) with cosine learning rate decay strategy. Weight decay and momentum are set to 0.0001 and 0.9, respectively.

**PASCAL VOC.** For PASCAL VOC benchmarks, we train our models on the VOC 2007 and 2012 *trainval* sets, which contain about $16,551$ images, and evaluate on the VOC 2007 *test* set (4952

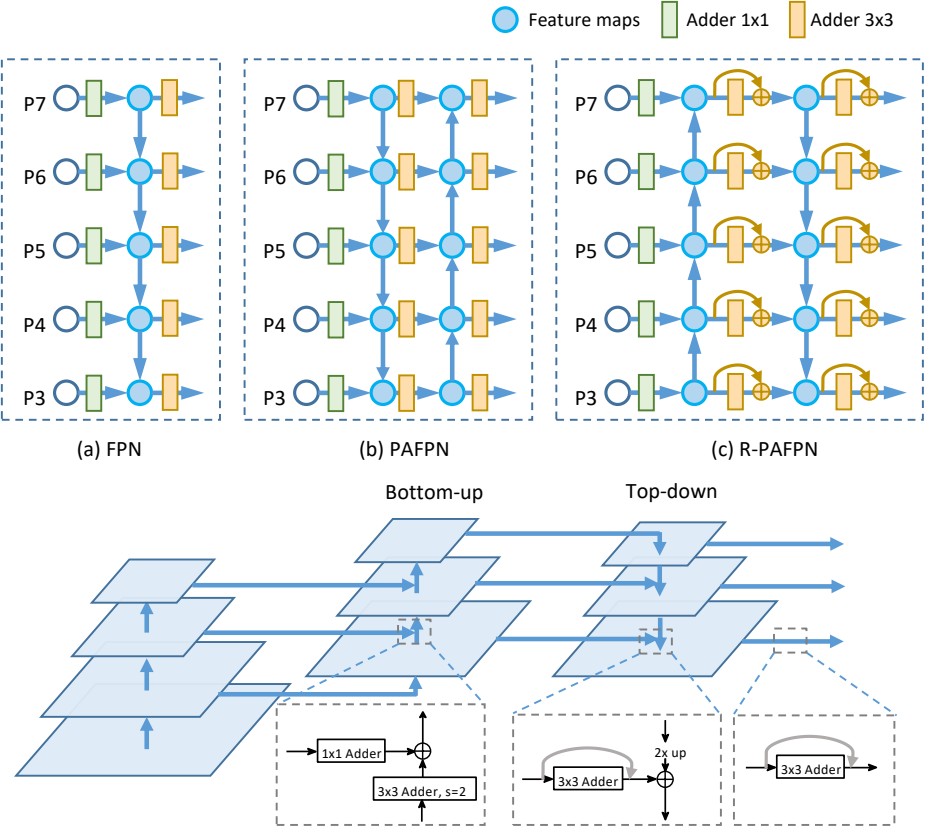

Figure 5: Sketches for different multi-scale feature fusion architectures. (a) Feature Pyramid Networks (FPN). (b) Path Aggregation Network (PAFPN). (c) The proposed R-PAFPN architecture for adder detectors. (d) More detailed architecture of R-PAFPN.

Table 1: Ablation studies for the baseline of adder detector. "NAN" indicates that the training is non-convergent. "LR" means using cosine learning rate and hyper-parameter tuning for learning rate.

| UnfrozenBN | $\ell_1$ Gradients | KD pre-trained | LR | mAP | $AP_{50}$ | $AP_{75}$ | $AP_S$ | $AP_M$ | $AP_L$ |
|---|---|---|---|---|---|---|---|---|---|
| | | | | NAN | - | - | - | - | - |
| ✓ | | | | 32.4 | 50.3 | 34.2 | 18.2 | 35.8 | 41.5 |
| ✓ | ✓ | | | 32.6 | 50.5 | 34.5 | 18.3 | 35.8 | 42.1 |
| ✓ | ✓ | ✓ | | 33.2 | 51.2 | 35.1 | 18.3 | 36.2 | 42.7 |
| ✓ | ✓ | ✓ | ✓ | **34.8** | **52.6** | **37.1** | **19.7** | **38.4** | **44.5** |

images). We use the VOC style mAP (*i.e.*,mAP at IoU=0.5) as the evaluation metric. The input images are resized to have shorter side being 600 while the longer side not to exceed 1000. Other hyper-parameters are similar to COCO benchmark. Our implementation is based on the popular object detection framework MMDetection [6].

## 4.2 Ablation Studies

**Steps towards a strong baseline.** We first analyze the strategies introduced in Section 3.1 on FCOS with adder backbone and neck. As shown in Table 1, if we train an adder FCOS detector with FrozenBN, the loss could not converge properly and the detector gets zero mAP. Updating the statistics of running mean and variance for BN (*i.e.*, Unfrozen BN) is critical for training the adder detector and achieve reasonable mAP of 32.4%. We further exploit $\ell_1$ gradients for input features instead of clipped $\ell_2$ gradients as in [5], which harvest about 0.2 mAP improvement.

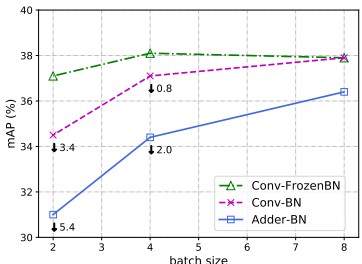

| Detector | Conv | bs=2 | bs=4 | bs=8 |
|----------|------|------|------|------|
| VFNet [48] | 44.5 | 34.6 | 40.5 | 41.5 |
| ATSS [49] | 39.6 | 29.3 | 34.5 | 36.9 |
| RepPoints [46] | 37.1 | 27.2 | 33.0 | 34.7 |
| RetinaNet [20] | 36.5 | 30.3 | 33.8 | 34.1 |
| GFL [18] | 40.2 | 31.4 | 35.8 | 37.9 |
| FoveaBox [15] | 36.5 | 31.9 | 33.8 | 34.7 |
| Sparse R-CNN [35] | 37.9 | 31.1 | 34.4 | 37.0 |

Figure 6: Comparisons of conv and adder detectors with various batch sizes.

Table 3: Performance on COCO *val2017* for different adder detectors with different batch sizes (bs). The second column shows the results of convolutional baselines.

To further improve the performance of the adder detector, we instead use a better pre-trained backbone, which is trained on ImageNet via kernel based knowledge distillation [45]. This strategy boosts the mAP to 33.2%. We also introduce cosine learning rate and hyper-parameter tuning for learning rate, which obtain 0.6% mAP improvement. Now we have got a strong baseline for adder detector, which achieves 34.8% mAP on COCO *val2017* set.

The AdderNet-50 trained with knowledge distillation [45] has 76.8% top-1 accuracy on ImageNet, which is slightly higher than that of ResNet-50. To explore to what extent the pre-trained backbones affect the final detection performance, we evaluate the backbones with and without knowledge distillation, as shown in Table 2. Using a vanilla AdderNet-50 with 1.9% top-1 accuracy drop on ImageNet suffers only 0.4% mAP drop compared with the KD counterpart. The AdderNet-18 trained with KD and the vanilla AdderNet-101 perform worse than their convolutional counterparts on ImageNet but still obtain considerably good performance for object detection.

**Impacts of batch size.** As discussed in Section 3.1.1, it is necessary to recompute the statistics of batch normalization layers when training the adder detectors. We first explore the performance of convolutional detectors with different batch sizes. As shown in Figure 6, when the batch size is relatively large enough, *i.e.*, batch size of 8, the performance of using FrozenBN and normal BN are similar. Reducing the batch size to 4 or 2, the accuracies of using BN

Table 2: Ablation studies for different pre-trained backbones. The values before the slash are Top-1 accuracy (%) on ImageNet and values after the slash are mAP on COCO for FCOS with corresponding backbones.

| | Top-1 Acc. (%) / Detection mAP (%) | | |
|---------|------|------|------|
| Backbone | R-18 | R-50 | R-101 |
| Conv | 69.8 / 35.2 | 76.2 / 39.0 | 77.37 / 41.0 |
| Adder | 67.0 / 33.3 | 74.9 / 37.4 | 76.08 / 38.9 |
| Adder KD | 68.8 / 33.9 | 76.8 / 37.8 | - |

(*i.e.*, recomputing statistics) decrease drastically while the counterpart with FrozenBN suffers slight performance degradation. Similar results are observed for adder detectors with normal BN, which demonstrates the intuitive fact that training detectors when recomputing statistics for BN with small batch size is challenging. Moreover, adder detectors suffer severer accuracy drop when reducing the batch size, *e.g.*, 5.4 mAP drop when decreasing batch size from 8 to 2 for adder detector while only 3.4 mAP drop for convolutional one. We perform more analysis on several modern detectors, as shown in Table 3. We replace the backbone of detectors with adder networks and explore the performance with different batch sizes. For most detectors, larger batch size consistently improves the performance, which demonstrates that large batch size is critical for training adder detectors.

**Impacts of different neck structures.** In Section 3.2, we propose a novel multi-scale feature fusion architecture (R-PAFPN) for improving the performance of adder detectors. Here we elaborate the effectiveness of the proposed design choices. Original FCOS [39] adopts feature pyramid network (FPN) for feature aggregation. Its adder counterpart achieves mAP of 34.8%, as shown in Table 4. We simply replace the neck architecture to PAFPN [22], which adds an extra bottom-up feature fusion path to FPN. This modification only brings minor improvement, *i.e.*,0.1% mAP, which demonstrates that simply exploiting more fusion paths is not enough for adder detectors and special design is urgent.

We further add extra shortcut connections to PAFPN (denoted as PAFPN w/ shortcut), motivated by the discussions in AdderSR [34]. Experiments show that adding skip connections is also beneficial

Table 4: Ablation studies for neck structures. The proposed R-PAFPN neck structure achieves the mAP of 36.5% on COCO *val2017* set.

| Neck | Type | mAP | AP$_{50}$ | AP$_{75}$ | AP$_S$ | AP$_M$ | AP$_L$ |
|---|---|---|---|---|---|---|---|
| FPN [19] | Adder | 34.8 | 52.6 | 37.1 | 19.7 | 38.4 | 44.5 |
| PAFPN [22] | Adder | 34.9 (+0.1) | 52.4 | 37.2 | 20.1 | 38.3 | 44.5 |
| PAFPN w/ shortcut | Adder | 36.1 (+1.3) | 53.8 | 38.9 | 20.4 | 39.8 | 46.5 |
| **R-PAFPN** | Adder | **37.0** (+2.2) | **55.5** | **39.7** | **21.7** | **40.5** | **47.4** |

for detection task, which bring 1.3% mAP improvement. As discussed in Section 3.2, the pre-trained backbone for adder detector exhibits sparser feature maps for deeper layers. Therefore, we propose to utilized bottom-up feature aggregation path to fuse features of different scales from the backbone network. As shown in Table 4, the proposed R-PAFPN neck structure harvests additional 0.9% mAP gains, achieving the mAP of 37.0% on COCO *val2017* set. It justifies our motivation that bottom-up path should be applied first since the low-level features could compensate for the sparse high-level features.

## 4.3 Experiments on COCO

We evaluate our proposed method on COCO benchmark for several state-of-the-art object detectors, including FCOS [39], FoveaBox [15], RetinaNet [20] and Sparse R-CNN [35]. We replace various components for these detection frameworks, *e.g.*, Backbone (B), Neck (N) and Head (H), as shown in Table 5. For example, FCOS with adder backbone and neck is denoted as Adder FCOS (B+H). Similar to prior method [34], we calculate the energy costs of different detectors. As discussed in prior literature [8, 47], one operation of floating-point addition and multiplication have energy costs of 0.9 $pJ$ and 3.7 $pJ$, respectively.

For standard $1\times$ schedule (*i.e.*, 12 epochs), our adder detectors with AdderNet-50 backbone and convolutional neck, achieve comparable performance on COCO *val2017* set with their convolutional baselines. For example, Adder FCOS (B) achieves mAP of 37.2% , which is only 1.2% mAP lower than FCOS [39]. Since the backbone is built with adder filters which get rid of massive multiplications, Adder FCOS (B) has considerably fewer number of multiplications than FCOS (129.9 vs. 214.7). Sparse R-CNN [35] with adder backbone only suffers 0.9% mAP drop while reducing the potential energy costs from 719.5 to 482. We also build a detector with adder backbone and neck, *i.e.*, Adder FCOS (B+N), which achieves 37.0% mAP and have further fewer multiplications. Compared with RetinaNet [20], our adder FCOS achieves the same detection accuracy but have much higher potential for energy efficiency.

We also conduct experiments on longer schedule (*i.e.*, $2\times$). Adder FCOS (B+N) achieves 37.8% mAP and outperforms RetinaNet [20] by 0.4% mAP. It obtains quite competitive performance with FoveaBox [15], Faster R-CNN [32] and RepPoints [46] but have much fewer multiplications. For ResNet-101 backbone, the adder counterpart suffers from 2.4% mAP drop while having $1.6\times$ energy reduction.

We further evaluated our method on two detectors with smaller input sizes so that they can be trained with larger batch size. Specifically, FCOS-RT [40] is a real-time version of FCOS, with the input image of $736 \times 512$, multi-scale training and longer training iterations (*i.e.*, 48 epochs). RetinaNet-MS-640 is

Table 6: Comparisons of mAP on PASCAL VOC.

| Model | Backbone | Neck | mAP |
|---|---|---|---|
| Faster R-CNN [32] | Conv R50 | Conv | 79.5 |
| FCOS [39] | Conv R50 | Conv | 79.1 |
| RetinaNet [20] | Conv R50 | Conv | 77.3 |
| FoveaBox [15] | Conv R50 | Conv | 76.6 |
| **Adder FCOS (Ours)** | Adder R50 | Adder | 76.5 |

trained with similar settings as [11], with $640 \times 640$ input images and also multi-scale training strategy. Adder FCOS-RT (B+N) achieves 1.3% mAP less that FCOS-RT while reducing the energy cost from 344.4 to 244.4. We try to replace all layers except the first and last layers in RetinaNet-MS-640 and obtain 35.2% mAP, which is 2.7% lower that the convolutional baseline but having $2\times$ energy cost reduction. Similarly, Adder RetinaNet-MS-500 achieves 3.0% mAP lower that the convolutional baseline with $2\times$ fewer energy cost. More comparisons with state-of-the-art detectors are shown in Figure 1. The proposed adder detectors present better accuracy and energy cost trade-off for various detection frameworks.

Table 5: Comparisons of object detection results on COCO *val2017*. We estimate the energy costs according to prior literature [8, 47], *i.e.*,one operation of floating-point addition and multiplication have energy costs of 0.9 $pJ$ and 3.7 $pJ$, respectively. B: Backbone, N: Neck, H: Head.

| Detectors | B | Adder? B | N | H | Epoch | #Mul (G) | #Add (G) | Energy (mJ) | AP$_{val}$ (%) |
|---|---|---|---|---|---|---|---|---|---|
| GHM [17] | R50 | | | | 12 | 250.3 | 250.3 | 1152 | 37.0 |
| PAFPN [22] | R50 | | | | 12 | 241.7 | 241.7 | 1112 | 37.5 |
| RetinaNet [20] | R50 | | | | 12 | 239.3 | 239.3 | 1100 | 36.5 |
| Libra R-CNN [28] | R50 | | | | 12 | 216.9 | 216.9 | 997.9 | 38.3 |
| Faster R-CNN [32] | R50 | | | | 12 | 215.8 | 215.8 | 992.8 | 37.4 |
| PISA [2] | R50 | | | | 12 | 215.8 | 215.8 | 992.8 | 38.4 |
| FSAF [51] | R50 | | | | 12 | 215.8 | 215.8 | 992.8 | 37.4 |
| RepPoints [46] | R50 | | | | 12 | 199.0 | 199.0 | 915.4 | 38.1 |
| FoveaBox [15] | R50 | | | | 12 | 215.8 | 215.8 | 992.7 | 36.5 |
| Adder FoveaBox | R50 | ✓ | | | 12 | 131 | 300.6 | 755.2 (0.76×) | 34.7 (-1.8) |
| Adder FoveaBox | R50 | ✓ | ✓ | | 12 | 112.9 | 318.7 | 704.4 (0.71×) | 33.3 (-3.2) |
| Sparse R-CNN [35] | R50 | | | | 12 | 156.4 | 156.4 | 719.5 | 37.9 |
| Adder Sparse R-CNN | R50 | ✓ | | | 12 | 71.59 | 241.2 | 482.0 (0.67×) | 37.0 (-0.9) |
| FCOS [39] | R50 | | | | 12 | 214.7 | 214.7 | 987.7 | 38.4 |
| Adder FCOS | R50 | ✓ | | | 12 | 129.9 | 299.5 | 750.2 (0.76×) | 37.2 (-1.2) |
| Adder FCOS | R50 | ✓ | ✓ | | 12 | 112.9 | 316.5 | 702.7 (0.71×) | 37.0 (-1.4) |
| RetinaNet [20] | R50 | | | | 24 | 239.3 | 239.3 | 1100 | 37.4 |
| Faster R-CNN [32] | R50 | | | | 24 | 215.8 | 215.8 | 992.8 | 38.4 |
| RepPoints [46] | R50 | | | | 24 | 199.0 | 199.0 | 915.4 | 38.6 |
| FoveaBox [15] | R50 | | | | 24 | 215.8 | 215.8 | 992.7 | 37.9 |
| Adder FoveaBox | R50 | ✓ | | | 24 | 131 | 300.6 | 755.2 (0.76×) | 35.8 (-2.1) |
| FCOS [39] | R50 | | | | 24 | 214.7 | 214.7 | 987.7 | 39.0 |
| Adder FCOS | R50 | ✓ | | | 24 | 129.9 | 299.5 | 750.2 (0.76×) | 38.2 (-0.8) |
| Adder FCOS | R50 | ✓ | ✓ | | 24 | 112.9 | 316.5 | 702.7 (0.71×) | 37.8 (-1.2) |
| FCOS | R18 | | | | 24 | 162.7 | 162.7 | 748.5 | 35.2 |
| Adder FCOS | R18 | ✓ | | | 24 | 141.9 | 183.5 | 690.3 (0.92×) | 33.9 (-1.3) |
| Adder FCOS | R18 | ✓ | ✓ | | 24 | 112.4 | 213.0 | 607.6 (0.81×) | 33.6 (-1.6) |
| FCOS | R101 | | | | 24 | 294.6 | 294.6 | 1355 | 41 |
| Adder FCOS | R101 | ✓ | | | 24 | 145.7 | 443.5 | 938.1 (0.69×) | 38.9 (-2.1) |
| Adder FCOS | R101 | ✓ | ✓ | | 24 | 113.3 | 475.9 | 847.4 (0.63×) | 38.6 (-2.4) |
| FCOS-RT [40] | R50 | | | | 48 | 74.86 | 74.86 | 344.4 | 40 |
| Adder FCOS-RT | R50 | ✓ | ✓ | | 48 | 39.16 | 110.6 | 244.4 (0.71×) | 38.7 (-1.3) |
| RetinaNet-MS-500 | R50 | | | | 50 | 61.24 | 61.24 | 281.7 | 35.3 |
| Adder RetinaNet-MS-500 | R50 | ✓ | ✓ | | 50 | 36.08 | 86.40 | 211.3 (0.75×) | 33.2 (-2.1) |
| Adder RetinaNet-MS-500 | R50 | ✓ | ✓ | ✓ | 50 | 10.34 | 112.1 | 139.2 (0.49×) | 32.3 (-3.0) |
| RetinaNet-MS-640 | R50 | | | | 50 | 95.68 | 95.68 | 440.1 | 37.9 |
| Adder RetinaNet-MS-640 | R50 | ✓ | ✓ | | 50 | 55.13 | 136.2 | 326.6 (0.74×) | 36.6 (-1.3) |
| Adder RetinaNet-MS-640 | R50 | ✓ | ✓ | ✓ | 50 | 16.15 | 175.2 | 217.4 (0.49×) | 35.2 (-2.7) |

Figure 7 shows some qualitative results of our proposed adder detectors and state-of-the-arts detectors, including RetinaNet [20] and FCOS [39]. Adder FCOS works well for a variety of challenging scenarios and have similar predictions with other detectors.

## 4.4 Experiments on PASCAL VOC

We also evaluate our proposed method on PASCAL VOC dataset. Our Adder FCOS with adder backbone and neck structure achieves mAP of 76.5, which is comparable with FoveaBox [15] and RetinaNet [20]. The performance is a bit poorer than Faster R-CNN and FCOS. However, considering the energy cost reduction, the proposed adder detector is a good trade-off for object detection accuracy and energy efficiency.

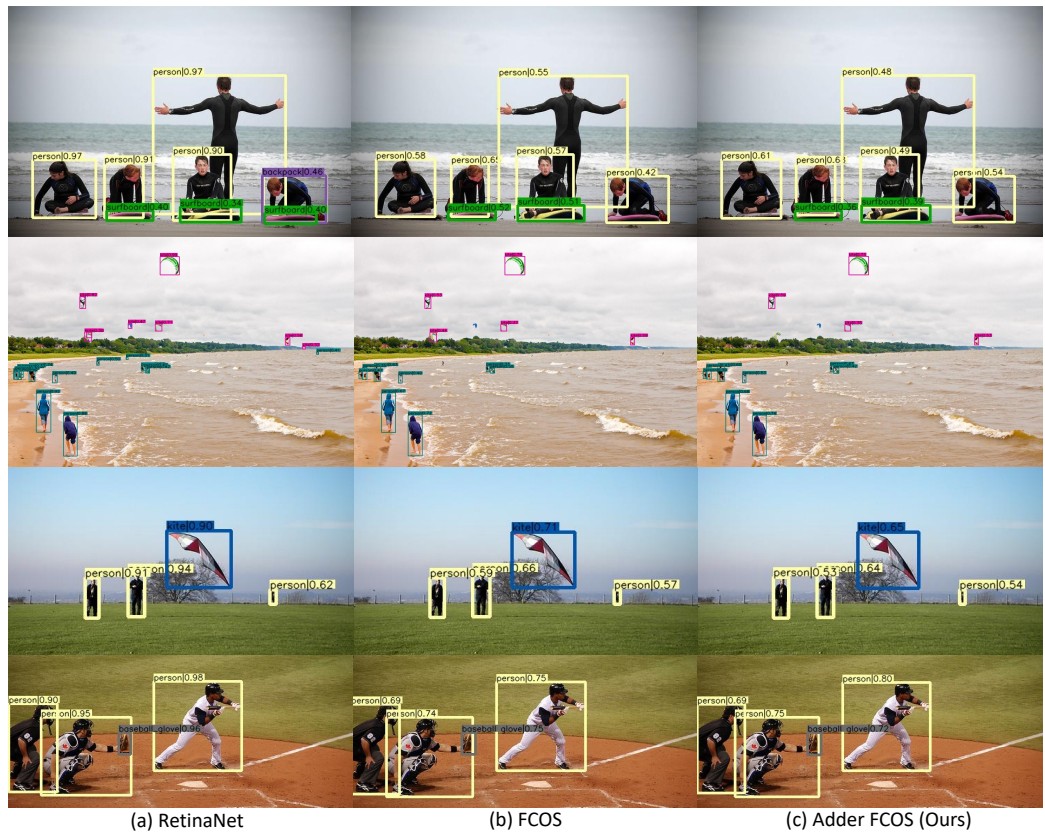

|  (a) RetinaNet | (b) FCOS | (c) Adder FCOS (Ours) |

Figure 7: Qualitative results of RetinaNet [20], FCOS [39] and the Adder FCOS.

## 5 Conclusion and Discussion

In this paper, we present an empirical study for accurate object detectors via adder neural networks. We first reveal that unfreezing statistics of batch normalization in backbone is crucial for adder detectors. We empirically analyze the properties of batch normalization and the impact of batch size. We also move an extra step forward to exploit a better architecture for adder detector. More specifically, we propose a novel reverse multi-scale feature fusion module called R-PAFPN, which compensates for the sparse high-level features with feature aggregation. Extensive experiments are conducted on COCO and PASCAL VOC benchmarks. In details, Adder FCOS achieved 37.8% AP on COCO val set, demonstrating comparable performance with convolutional counterpart but having much more potential energy reduction.

There are still some unsolved limitations for adder detector. For example, it still suffers from a bit accuracy degradation when compared with its convolutional counterpart, *i.e.*, FCOS [39]. What's more, the prediction heads on some adder detectors are still stacked with convolutions, and we empirically find that it would bring considerably large performance reduction if we replace convolutional filters on heads with adder filters for FCOS. Nevertheless, considering the energy cost saving brought by AdderNets, it is still a promising solution for efficient object detectors. In further work, it would be interesting to design a new prediction head architecture which is suitable for accurate adder detectors. We hope this study will be helpful for the research of adder neural networks and energy-efficient object detection.

## Acknowledgments and Disclosure of Funding

The authors sincerely thank anonymous reviewers and ACs for their helpful comments. Chang Xu was supported in part by the Australian Research Council under Projects DE180101438 and DP210101859.

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
