# Supplementary Materials: An Empirical Study of Adder Neural Networks for Object Detection

**Xinghao Chen[1], Chang Xu[2], Minjing Dong[2,1], Chunjing Xu[1], Yunhe Wang[1]** *

[1]Huawei Noah's Ark Lab
[2]School of Computer Science, University of Sydney
{xinghao.chen, yunhe.wang}@huawei.com, c.xu@sydney.edu.au

## A   Quantization

As discussed in prior literature [1, 4], one operation of floating-point addition and multiplication have energy costs of 0.9 $pJ$ and 3.7 $pJ$, respectively. Meanwhile, one operation of 8-bit integer addition and multiplication have 0.03 $pJ$ and 0.2 $pJ$ energy costs, demonstrating much lower cost than floating-point operation. Therefore, it is important to explore whether adder detectors performs well for INT8 quantization. We tried to adopt INT8 post quantization for our Adder FCOS (B+N) model, which suffers 0.8 mAP drop compared with full precision model, as shown in Table A. The energy reduction further increases from 29% to 35%. Note that post training quantization is not optimal for INT8 models, and quantization-aware training may greatly further improve the accuracy.

Table A: Quantitative results of INT8 convolutional and adder models

|  | #MUL | #ADD | FP32 | | INT8 | |
|---|---|---|---|---|---|---|
|  |  |  | Energy | mAP | Energy | mAP |
| FCOS | 214.7 | 214.7 | 987.62 | 38.4 | 49.38 | 38.1 |
| Adder FCOS (B+N) | 112.9 | 316.5 | 702.58 (0.71×) | 37.0 (-1.4) | 32.08 (0.65×) | 36.2 (-1.9) |

## B   Training Tricks for CNN-based Detectors

The training tricks in Table 1 in the main body of this paper mainly include well-tuned learning rate with cosine decay. We also tried to utilize these tricks for training CNN-based object detectors. As shown in Table B, these tricks bring 0.2%-0.6% mAP gain for various CNN-based detectors. On contrast, this strategy improves the adder detector for 1.2% mAP, which indicates that the well-developed hyper-parameters for CNN-based detectors are often not optimal for adder detectors.

Table B: Comparing CNN-based detectors and adder detectors for learning rate strategy.

| Model | Step LR | Cosine LR + LR tuning |
|---|---|---|
| RetinaNet | 36.4 | 36.6 (+0.2) |
| FCOS | 38.4 | 39.0 (+0.6) |
| Adder FCOS | 33.2 | 34.8 (+1.2) |

## C   Robustness to Domain Shift

It is an interesting topic to explore the robustness to the domain shift for AdderNet-based detector. We utilize the FCOS models trained on COCO to evaluate on the Cityscapes dataset for the common

---

*Corresponding author.

35th Conference on Neural Information Processing Systems (NeurIPS 2021).

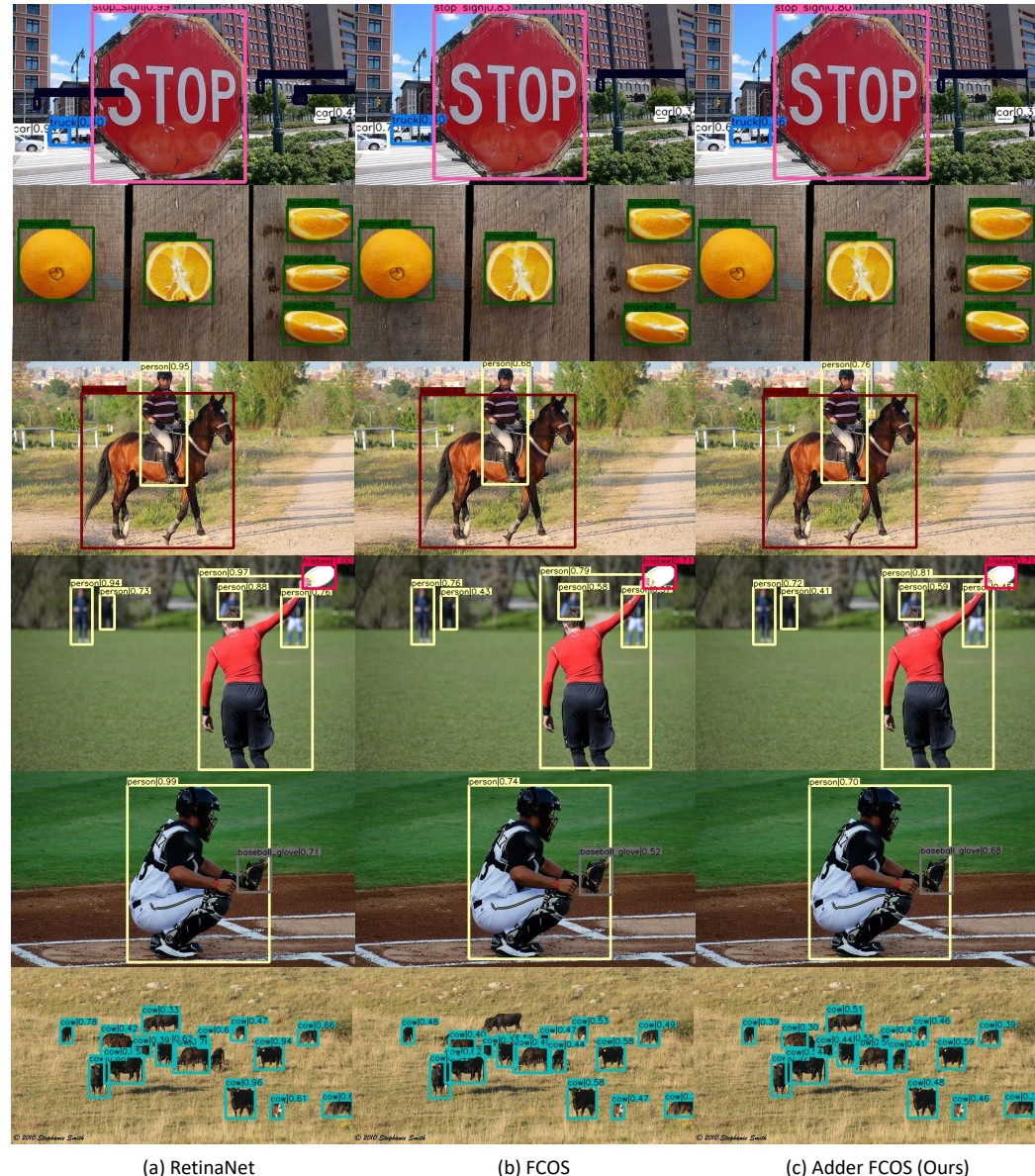

| (a) RetinaNet | (b) FCOS | (c) Adder FCOS (Ours) |

Figure 1: Qualitative results of RetinaNet [2], FCOS [3] and our proposed Adder FCOS.

object categories without fine-tuning. More specifically, we adopt the coco-trained models to predict the detection results for Cityscapes val set, and map the original ground-truth categories of Cityscapes to COCO (80 classes) to calculate the mAP. As shown in Table C, Adder FCOS suffers from 2.2% mAP drop on Cityscapes compared with convolutional counterpart, which is similar with the performance drop on COCO. This demonstrates that adder detectors have similar robustness performance to the domain shift with CNN detectors.

Table C: Robustness to the domain shift for AdderNet-based and CNN-based detectors.

| Model | COCO mAP | Cityscapes mAP |
| --- | --- | --- |
| FCOS | 38.4 | 29.4 |
| Adder FCOS (B+N) | 36.5 (-1.9) | 27.2 (-2.2) |

## D  Visualization

Figure 1 shows more qualitative results of our proposed adder detection and state-of-the-arts detectors, including RetinaNet [2] and FCOS [3]. Adder FCOS works well for a variety of challenging scenarios and has similar predictions with other detectors. Qualitative results for RetinaNet-MS-600 and its adder variants are shown in Figure 2.

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

Figure 2: Qualitative results of RetinaNet-MS-640 [2] and our proposed adder detectors.