# OpenReview forum: "An Empirical Study of Adder Neural Networks for Object Detection"
_NeurIPS.cc/2021/Conference — NeurIPS 2021 Poster_

### Official Review · Reviewer_h7uh · 2021-07-14

**Rating:** 5
**Confidence:** 4

**Summary:**

The authors look at using AdderNets for object detection. AdderNets are neural networks which replace dot product between a weight filter and an activation window with the L1 distance between that weight and activation window instead. In hardware, AdderNets would replace multiplications with additions which are simpler and use less energy.

The authors use empirical experiments to discover a number of strategies that improve AdderNet AP, including:
1. Unfreezing the batchnorm statistics during downstream fine-tuning
2. Using direct gradients for the L1 distance (which are +`1/-1) instead of a continuous approximate gradient
3. Finetuning from a knowledge-distilled AdderNet backbone (as opposed to a regularly trained AdderNet)
4. Architectural changes specific to object detection, adding more skip connections

With these ideas, AdderNet versions of object detectors can approach 1-3% of their convolutional counterparts in AP while reducing multiplies by about 50% and energy consumption by `about 25%.

**Limitations And Societal Impact:**

Paper topic is efficient ML, will help reduce energy footprint of DNN inference for object detection.

**Main Review:**

Strengths:
 - Clearly written paper with practical ideas that seem easy to apply to real object detection systems
 - Large amount of empirical data validating claims, Table 5 especially shows that their technique achieves similar results across a large variety of baseline CNN object detectors. I think the paper is very trustworthy.

Weaknesses:
 - Impact of each individual tweak seems small. Other than unfreezing the BN, the other ideas only seem to boost AP by a small amount (Table 1). This amount seems comparable to what a practitioner would get by doing hyperparameter tuning, cleaning data, or some other generic tuning.
 - Energy savings seem underwhelming. Table 5 seems to show that we can save 25% energy at a cost of 1-3% loss of AP. That doesn't seem much better than using a 25% shallower convnet. I'm a bit surprised - I would have thought converting all multiplies in the convs to adds would save a lot more.

One way to improve the paper: right now the energy calculation assumes floating-point adds/muls. What about comparing fixed-point (e.g. 16-bit) operations instead? Are there other benefits to AdderNets (throughput, latency, faster training) that could be leveraged to make the results more impressive?


**Time Spent Reviewing:**

1

---

> ### Author Response · Authors · 2021-08-10
> **Response to Reviewer h7uh**
>
> We thank the reviewer for the constructive comments.
>
> **Impact of each individual tweak seems small**
>
> Thanks for the concerns. The results shown in Table 1 indicate the different components do have considerably significant improvements. Other techniques like hyper-parameter tuning, cleaning data, or some other generic tuning may also further be beneficial for the adder detectors.
>
> **Energy savings seem underwhelming**
>
> Replacing only the backbone with adder typically leads to 25%-30% energy reduction as the FLOPs of backbone occupies 40%-50% of the whole detectors. The energy reduction can reach 50% if all components of detector, i.e. backbone, neck and head, are replaced with adder filters, as shown in the last row in Table 5. We will add a column to Table 5 to clearly indicate the percentage of energy reduction.
> What is more, we have compared with shallower convolutional networks. In Figure 1, the three dots for FCOS/Adder FCOS represent the results of using R-18/R-50/R-101 backbones. We can see that Adder FCOS with R-101 achieves similar accuracy with Conv FCOS with R-50, while having much lower energy cost. We will include more details in the final version.
>
> **Fixed-point operations**
>
> Thanks for the good suggestions. We have tried to adopt INT8 post quantization for our adder model, which suffers only 0.4 mAP drop compared with full precision model. The energy reduction further increases to 35%. We will include more results for INT8 models in the final version. What is more, quantization aware training may further improve the accuracy of INT8 models.
>
> |                        |    #MUL     |    #ADD     |    FP32 Energy    |    INT8 Energy    |
> |------------------------|-------------|-------------|-------------------|-------------------|
> | FCOS                   |    214.7    |    214.7    |    987.62         |    49.381         |
> |    Adder FCOS (B+N)    |    112.9    |    316.5    |    702.58         |    32.075         |
> |    Energy reduction    |             |             |    29%            |    35%            |

---

### Official Review · Reviewer_ZgMW · 2021-07-15

**Rating:** 5
**Confidence:** 4

**Summary:**

This paper presents an empirical study for exploiting the AdderNet for low-cost object detection including the considerations about batch normalization, feature pyramid networks. Experiments have demonstrated that the proposed Adder FCOS can achieve competitive performance with less energy cost.

**Limitations And Societal Impact:**

Yes. The authors have discussed partial of the limitations of the proposed method.

Typos:
* L259: Adder FCOS (B+H) -> Adder FCOS (B+N)


**Main Review:**

 Strength:
1. This paper is well-organized and easy to understand.
2. The proposed Adder FCOS and other variants are effective for low-cost object detection and experimental results are good.

 Weakness:
1. The technical novelty of this paper is limited. The authors exploit the AdderNet to object detection approaches with some modifications (e.g., unfreezing Batch Normalization, better pretrained backbone, etc.) to stabilize the training and improve the performance.
2. In Sec. 2.2, the authors claim the sparse features caused by adder filters and propose a reversed feature pyramid network (R-PAFPN) to alleviate the problem while there no evidence to prove it. In Table 4, it seems that the shortcut connections matter more than the bottom-up features. Besides, providing the results of R-PAFPN w/o shortcut will make it more clear for us to determine whether using bottom-up features can alleviate the sparse problem. In addition, visualizations of features after the R-PAFPN are required for comparisons.
3. I’m concerned about whether Synchronized Batch Normalization is adopted in the implementation since small batch in one GPU (<8) will degrade the performance.
4. With longer training schedule, the performance gaps between Adder FCOS and FCOS become larger (as shown in Table 5). Is this a general limitation of the proposed approach?
5. How about the inference speed? Current computing bottlenecks of low-cost processors lie in memory access [1]. The Adder FCOS reduces the energy cost but still involves higher memory access cost, which might limit the performance in practical use.

[1] Ma et.al. ShuffleNet V2: Practical Guidelines for Efficient CNN Architecture Design.  ECCV 2018.


**Time Spent Reviewing:**

8

---

> ### Author Response · Authors · 2021-08-10
> **Response to Reviewer ZgMW**
>
> We thank the reviewer for the constructive comments.
>
> **The technical novelty of this paper is limited**
>
> This paper extends the success of AdderNet to the detection task by developing adder detectors. However, this extension is nontrivial as simply replacing convolutional filters in an object detectors with adder filters results in unsatisfying performance. In this paper we present an extensive empirical study for developing accurate adder detectors. A bag of strategies is carefully designed to upgrade object detectors with adder filters, including the role of batch normalization and a new feature fusion network. Extensive experiments on coco for various detectors demonstrate the effectiveness of the proposed method.
>
>
> **Concerns about R-PAFPN**
>
> Thanks for the suggestions. We have added the experiments for R-PAFPN without shortcut. Experimental results show that R-PAFPN without shortcut achieve mAP of 35.4, which is 1.1% mAP lower than the R-PAFPN and 0.5% mAP higher than PAFPN. Since the only difference between R-PAFPN w/o shortcut and PAFPN is the proposed feature fusion module, it also demonstrates that the proposed module can bring 0.5% mAP gain.
>
> **SyncBN**
>
> Thanks for the comments and all experiments are performed without SyncBN. We have done some experiments for the impact of batch size in P6 Line 223 and Figure 6. For convolutional detectors, batch size of 4 or 8 have already achieved considerably good performance. However, larger batch size has become more critical for adder detectors, which may also attribute to the unstable BN statistic as discussed in Section 2.1.1. We have also tried to adopt SyncBN for adder detectors. However, we observe performance drop for various detectors. The main reason may also due to the unstable BN statistic and we will include more discussions in the final version.
>
> **The performance gaps between Adder FCOS and FCOS with longer training schedule**
>
> From Table 5, we can see that for 12 training epochs, the performance gap between Adder FCOS (B+N) and FCOS is 1.9 mAP. This gap has been reduced to only 1.2 mAP for longer training schedule, i.e. 24 epochs. Therefore, adder detectors may be beneficial from longer training schedule.
>
> **Inference Speed**
>
> Thanks for the nice concern. In general hardware like GPU and off-the-shell deep learning frameworks, adder networks do not exhibit faster inference speed due to the fact that the matrix multiplication has been highly optimized in CUDA and GPU, while the adder operations are not. However, some recent work have demonstrated that AdderNet can achieve 16% speedup in FPGA [R1] (see Section 4, P8) for ResNet-18. The proposed AdderNet-based detectors also utilize ResNet as backbone, thus they should also achieve some speedup. We also would like to emphasize that the main advantage of adder networks lies on the energy cost and chip area reduction on custom-tailored hardware, while AdderNets would not present huge inference speedup.
>
> [R1] AdderNet and its Minimalist Hardware Design for Energy-Efficient Artificial Intelligence, arXiv: 2101.10015, 2021
>
> **Typos**
>
> Thanks for pointing out the typos. We will fix all typos and proofread the manuscript carefully.

---

### Official Review · Reviewer_uAeA · 2021-07-16

**Rating:** 4
**Confidence:** 4

**Summary:**

The purpose of the paper is to utilize addernet to replace the convolutional layer in the object detection network, so as to optimize the power consumption of the model. To address the problem of directly replacing the convolutional layer with the addernet, the author has made two contributions: (1) When training the addernet, the BN layers in the pre-trained weights are not frozen (2) Considering the sparse nature of the adder network, the paper designed a new FPN to alleviates the problem.

**Limitations And Societal Impact:**

No obvious limitations and potential negative societal impact.

**Main Review:**

- The paper is well-written and the authors introduced the contribution and related work in detail, and verified the generalization on many detectors.
- The innovation of this article is relatively lacking. Among the two contributions of this article, bn-no-freeze during training is more like a training trick. The second contribution of this article is to redesign the FPN for the sparsity of addernet. Although the paper compares the feature sparsity of normal convolution and addernet, it does not show the effect of R-PAFPN on normal convolution experimentally. And the largest gain part is on the shortcut, which obviously increases the amount of calculation of the network, which makes the design of R-PAFPN more like an incremental improvement to FPN. In general, although the contribution of this article presents a certain technical contribution, the novelty of the paper to the NIPS community may be relatively small.
-  There are some unfair setting between the Conv and Adder detectors. For example, distillation is used in Pre-training stage of adder network, but the pre-trained weight of the Conv network does not have distillation.
- There is no comparison with the performance of Conv network when using R-PAFPN.
- For the power-efficient design, there is no comparison over model pruning to show whether addernet has a better power-accuracy trade-off. Besides, there are certain doubts whether addernet and prune can coexist well as there is no discussion and experiments.

**Time Spent Reviewing:**

2

---

> ### Author Response · Authors · 2021-08-10
> **Response to Reviewer uAeA**
>
> We thank the reviewer for the constructive comments.
>
> **The innovation of this article is relatively lacking**
>
> Thanks for the concerns. This paper extends the success of AdderNet to the detection task by developing adder detectors. However, this extension is nontrivial as simply replacing convolutional filters in an object detectors with adder filters results in unsatisfying performance. In this paper we present an extensive empirical study for developing accurate adder detectors. A bag of strategies is carefully designed to upgrade object detectors with adder filters, including the role of batch normalization and a new feature fusion network. Extensive experiments on coco for various detectors demonstrate the effectiveness of the proposed method.
>
> 1) We would like to emphasize that utilizing unfrozen BN for adder detector is not a trivial training trick. We provide extensive discussions and insights that it is essential to unfreeze the statistic of batch normalization when fine-tuning the adder detectors from a pre-trained backbone.
> 2) We apply R-PAFPN for convolutional detectors and observe that R-PAFPN achieves similar accuracy with PAFPN, which demonstrates that the design of R-PAFPN is essential for adder detectors and justifies our motivation.
> 3) The additional shortcut only introduce very few computation overhead, e.g. only 11M Add for FCOS, which is quite insignificant when compared with about 300G Add for the whole network.
>
> **Unfair setting between the Conv and Adder detectors**
>
> We use KD pre-rained backbone for adder network mainly due to low performance of vanilla Adder R-50, i.e., 74.9% top-1 accuracy. In fact, the accuracy of KD retrained Adder R-50 (76.8%) is on par with the standard R-50 (76.2%), which is a nearly fair comparison. We also provide ablation experiments for different pre-trained backbones in Table 2. With vanilla Adder R-50 (74.9% top-1), the detection performance only has 0.4 mAP drop.
>
> **Comparison with the performance of Conv network when using R-PAFPN**
>
> FCOS with R-PAFPN achieves mAP of 38.8%, which is 0.4% higher than FCOS with FPN and similar with PAFPN. We will include this result in the final version.
>
> **Comparison with pruning method**
>
> Thanks for the good suggestion. We compare with the pruning results in Table 2 of Joint-DetNAS (Yao et al., CVPR 2021). When pruning 30% channels for R50 backbone (i.e. 221.02 #MUL, 221.02 #ADD and 1017 mJ energy with only 8% energy reduction), the detector has 0.7 mAP drop. In contrast, various detectors with adder backbone has over 25% energy reduction while only suffering 0.8~2 mAP drop, demonstrating better energy-accuracy trade-off. What is more, AdderNet can apply for pruned architecture and channel pruning methods can also utilized for pruning AdderNet, which is an interesting topic for future direction.

---

### Official Review · Reviewer_NNwT · 2021-07-21

**Rating:** 6
**Confidence:** 5

**Summary:**

This paper empirically explores the impact of adder-based network architecture for developing object detectors. It studies the impact of batch normalization and batch size on the performance of adder networks and proposes unfreezing batch normalization for adder networks. It also explored the effect of different neck architectures by investigating the limitations of adder networks and how to improve them.

**Limitations And Societal Impact:**

A major limitation not properly addressed is the fact that in some cases the adder networks cannot be trained. Additional hyperparameter tuning would be necessary in this case beyond the standard procedures.

**Main Review:**

This paper proposes a series of strategies to reform object detectors with adder filters. Overall, it sheds more light into how to make adder networks work better for object detection. There are some points that should be incorporated into the paper.

- Section 2.1.2 needs to be elaborated more.

- The way Figure 6 and Table 3 are placed makes it confusing to read.

- In figure 3 the accuracy (mAP) should also be included. The lower loss may not always lead to hiegher mAP values.

- The energy savings ignore memory read and write operations which depend on feature map size and parameters and constitute a large part of the energy consumption hence, the fraction of improvement might not be that significant if all these factors are considered. Can you please elaborate more on this and add a relevant explanation in the paper.

**Time Spent Reviewing:**

2

---

> ### Author Response · Authors · 2021-08-10
> **Response to Reviewer NNwT**
>
> We thank the reviewer for the constructive comments.
>
> **Section 2.1.2 needs to be elaborated more**
>
> Thanks. We will include more details in Section 2.1.2 in the final version.
>
> **The way Figure 6 and Table 3 are placed makes it confusing to read**
>
> Sorry for the confusion. We will place Figure 6 and Table 3 in different rows to make them easier to read.
>
> **In figure 3 the accuracy (mAP) should also be included**
>
> Thanks for the suggestions. The corresponding mAPs are listed in the first two rows of Table 1. We will include the mAPs in the legends of Figure 3.
>
> **Discussions for memory read and write operations**
>
> Thanks for the good point. The only difference between convolution and adder filter is the way of calculating similarity of input feature and filter weight. Therefore, the memory read and write operations exhibit very similar energy consumption for CNNs and AdderNets, since the feature map size and parameters are the same. It is an interesting topic and we will include more discussions in the final version.
>
> **A major limitation not properly addressed is the fact that in some cases the adder networks cannot be trained**
>
> In the manuscript we provide extensive discussions and insights that with FrozenBN the adder detectors fail to converge. It is mainly due to the unstable feature statistic instead of training hyper-parameters. By using the proposed UnfrozenBN, the adder detectors can be trained to achieve an reasonable accuracy.

---

### Decision · Program_Chairs · 2021-09-28

**Decision:**

Accept (Poster)

**Comment:**

On the whole the reviewers agreed that the paper was well presented and thoroughly evaluated. There were some points raised on how much of an improvement this provides and some clarity questions, but these concerns were addressed by the authors in the rebuttal, and I think the paper results will be of interest to the NeurIPS community.

**Consistency Experiment:**

NeurIPS has a long history of experimentation. In 2014, NeurIPS ran an experiment in which 10% of submissions were reviewed by two independent committees to quantify the randomness in the review process. This year, we repeated a variant of this experiment to see how the quality of the review process has changed over time.  This paper was part of the experiment and was therefore assigned to two committees (consisting of reviewers, an Area Chair, and a Senior Area Chair) that reached independent decisions.  If both committees made the same recommendation, this recommendation was followed. If a single committee recommended acceptance, the paper was accepted (with the exception of a few cases in which the other committee identified what we considered a fatal flaw, e.g., an error in a key result).

This copy’s committee reached the following decision: **Accept (Poster)**

The other committee assigned to the paper recommended **Reject**.  You can find the other set of reviews, along with any follow up discussion with the authors here:
https://openreview.net/forum?id=_9oQ9pAYYX